# Phytosterol-Loaded Surface-Tailored Bioactive-Polymer Nanoparticles for Cancer Treatment: Optimization, In Vitro Cell Viability, Antioxidant Activity, and Stability Studies

**DOI:** 10.3390/gels8040219

**Published:** 2022-04-02

**Authors:** Shahid Karim, Md Habban Akhter, Abdulhadi S. Burzangi, Huda Alkreathy, Basma Alharthy, Sabna Kotta, Shadab Md, Md Abdur Rashid, Obaid Afzal, Abdulmalik S. A. Altamimi, Habibullah Khalilullah

**Affiliations:** 1Department of Pharmacology, Faculty of Medicine, King Abdulaziz University, Jeddah 21589, Saudi Arabia; skaled@kau.edu.sa (S.K.); burzangi@kau.edu.sa (A.S.B.); halkreathy@kau.edu.sa (H.A.); bmtalharthy@kau.edu.sa (B.A.); 2School of Pharmaceutical and Population Health Informatics (SoPPHI), DIT University, Dehradun 248009, India; 3Department of Pharmaceutics, Faculty of Pharmacy, King Abdulaziz University, Jeddah 21589, Saudi Arabia; skotta@kau.edu.sa (S.K.); shaque@kau.edu.sa (S.M.); 4Department of Pharmaceutics, College of Pharmacy, King Khalid University, Abha, Aseer 62529, Saudi Arabia; mdrashid@kku.edu.sa; 5Department of Pharmaceutical Chemistry, College of Pharmacy, Prince Sattam Bin Abdulaziz University, Al-Kharj 11942, Saudi Arabia; o.akram@psau.edu.sa (O.A.); as.altamimi@psau.edu.sa (A.S.A.A.); 6Department of Pharmaceutical Chemistry and Pharmacognosy, Unaizah College of Pharmacy, Qassim University, Unaizah 51911, Saudi Arabia; h.abdulaziz@qu.edu.sa

**Keywords:** breast cancer, chitosan, nanoparticles, folic acid, β-sitosterol, drug delivery

## Abstract

This study aimsto optimize, characterize, and assess the phytosterol-loaded surface-tailored bioactive Alginate/Chitosan NPs for antitumor efficacy against breast cancer. β-Sitosterol-loaded Alginate/Chitosan nanoparticles (β-SIT-Alg/Ch-NPs) were fabricated using an ion-gelation technique, and then the NPs’ surfaces were activated using an EDC/sulfo-NHS conjugation reaction. The activated chitosan NPs werefunctionalized with folic acid (FA), leveled as β-SIT-Alg/Ch-NPs-FA. Moreover, the functionalized NPs were characterized for size distribution, polydispersity index (PDI), and surface charge, FT-IR and DSC. β-SIT released from β-SIT-Alg/Ch-NPs was estimated in various biorelevant media of pH 7.4, 6.5, and 5.5, and data werefitted into various kinetic models. The cytotoxic study of β-SIT-Alg/Ch-NPs-FA against the cancer cell line was established. The antioxidant study of developed β-SIT-Alg/Ch-NPs was performed using DPPH assay. The stability of developed optimized formulation was assessed in phosphate buffer saline (PBS, pH 7.4), as per ICH guidelines. The drug-entrapped Alg/Ch-NPs-FA appeared uniform and nonaggregated, and the nanoscale particle measured a mean size of 126 ± 8.70 nm. The %drug encapsulation efficiency and %drug loading in β-SIT-Alg/Ch-NPs-FA were 91.06 ± 2.6% and 6.0 ± 0.52%, respectively. The surface charge on β-SIT-Alg/Ch-NPs-FA was measured as +25 mV. The maximum β-SIT release from β-SIT-Alg/Ch-NPs-FA was 71.50 ± 6.5% in pH 5.5. The cytotoxic assay expressed an extremely significant antitumor effect by β-SIT-Alg/Ch-NPs-FA when compared to β-SIT-suspension (*p* < 0.001). The antioxidant capacity of β-SIT-Alg/Ch-NPs-FA was 91 ± 5.99% compared to 29 ± 8.02% for β-SIT-suspension. The stability of NPs noticed an unworthy alteration (*p* > 0.05) in particle sizes and other parameters under study in the specific period.

## 1. Introduction

Breast cancer (BC) is the leading concern of death among women, with a mortality rate of higher than 1.6% and 1.1 million new cases of breast cancer diagnosed each year globally. Regarding new cases of 2021, out of the 281,550 diagnosed, 43,600 deaths took place in females in the United States alone due to breast cancer [1]. Cancer is a lethal disease that has multifactorial origins with uncontrolled cell division, which can evade any part of the body due to high proliferation, invasion, and metastasis. Breast cancer is common in women, attaining second spot after lung cancer, and is highly challenging to public concerns. The breast carcinoma could be ductal or lobular, based on the histopathological assessment. There are many cell-surface-receptor expressions seen on a molecular basis in breast tumor, viz. progesterone receptor (PR), epidermal growth factor receptor 2 (HER2), triple-negative receptor, and folate receptor [2]. A few decades back, various BC treatment strategies were used in the clinics. The conventional therapy outlines the chemotherapy, radiotherapy, or surgical removal, which remains inadequate in therapeutic modalities for eradication of the disease, and which is generally associated with high-dose-related toxicity, unspecific treatment, and cause severe unwanted side effects. However, chemotherapy is the most successful to date, but its nonselective delivery approach, serious adverse effects, and multidrug resistance limits its application [3].

Nanoparticle-based therapy has gained substantial popularity in recent decades and appears to be a feasible modern therapeutic option because of its specificity and selectivity, as well as its capacity to restrict off-target drug release, while raising drug concentration at the target area. Further, techniques towards drug targeting by several nanocarrier-based therapies have been explored, including inorganic nanoparticles, (e.g., magnetic nanoparticles, silica nanoparticles), organic nanocarriers (PLGA, PLA, polymer micelles, ferritin), lipidic nanocarriers (e.g., Liposomes), polymeric nanocarriers, dendrimers, carbon nanotubes, and fullerenes [4]. Nanocarriers can infiltrate the tumor vasculature through its porous endothelium and large fenestration, and disorient architecture of the tumor microenvironment via a process called enhanced permeation and retention (EPR) impact/passive targeting [5]. EPR has the potential to improve the nanoparticle accumulation surrounding the tumor region, although it has several limitations, which are to be addressed and considered critically in various preclinical models. The generation of hydrostatic pressure, interstitial fluid pressure, and osmotic pressure largely rely upon the tumor microenvironment associated with extravasation, poor lymph drainage, large fenestration, impaired endothelial cells, dense extracellular matrix, dense fibers, and aggressive angiogenic factors, which severely limit the inflow and outflow of small molecules to the target site, resulting in suboptimal drug chemotherapeutic delivery. In addition, the EPR-based pathway in cancer treatment depends upon the type of cancer, disease location, their pathophysiology, and xenograft models [6]. Paying attention to active drug targeting, which is mediated via recognition by the cell-surface receptor of ligated nanocarriers on the tumor cells and endothelium, is highly applicable to overcome the shortcomings faced on EPR-based therapy. Engineering nanocarriers with ligands, a protein molecule is an advanced and effective technology for specific targeting to cancer cells/tissue. Therefore, a novel strategy to ameliorate the targetability and efficacy of drugs to the solid tumor is a compelling indigence, and requires a critical understanding and exploration in the scientific arena [7,8].

Folic acid is an essential component for the synthesis of nitrogenous bases required for amino acid metabolism, and by this means controls and regulates cell growth, proliferation, and survival. The high level of folate-receptor expression in solid tumors could be considered as an ideal target for therapeutic delivery [9].

β-sitosterol is a plant-derived major phytosterol abundantly present in vegetable oils and dry fruits, and has received wider attention due to a broad spectrum of activity, including its effectiveness towards reducing intestinal absorption of cholesterol; reducing low-density lipoprotein, thus offering protection against cardiovascular aliments; and suppressing cancer growth. Studies reported that it has remarkable biological effects, including anti-inflammatory, chemopreventive, antioxidant, antibacterial, antifungal, wound-healing, antidiabetic, and antihypertensive effects [10,11].

The alginate molecules can be complexed stably with chitosan through electrostatic attraction, and then folic acid can be bicovalently conjugated to chitosan molecules via its γ-carboxyl moiety, and the resulting component retains a high affinity for breast cancer cells bearing the overexpressed folate receptor [12,13]. Chitosan and alginate work synergistically to protect entrapped active molecules from oxidation, enzymatic degradation, and hydrolysis [14]. Chitosan and alginate contain hydroxyl, amino, and carboxyl groups, which increase the solubility and residence time of hydrophobic molecules in a nanosystem. Encapsulation using alginates is most often carried out by the dispersion of the alginate/encapsulant solution onto a medium of calcium chloride. Contact between the alginate and the calcium in solution induces immediate interfacial ionic polymerization of the alginate via binding of calcium ion within the cavities of the guluronic residues, thus forming a polyanionic nanoparticle [15]. The objective of this work was to formulate a β-Sitosterol-loaded surface-tailored nanocarrier for drug targeting to BC. To improve the stability and targeting effect of the Alg/Ch-NPs, folic acid, a ligand for folate receptor, was conjugated to β-Sitosterol-loaded Alg/Ch-NPs.

## 2. Results and Discussion

### 2.1. Optimization

The formulation of chitosan/alginate NPs was statistically optimized to apply3^3^-BBD within the stipulated time. The effect of formulation variables chitosan (X1), sodium alginate (X2), and calcium chloride (X3) concentration, used in three levels—low (−1), medium (0), and high (+1)—were studied on the following responses: particle size (nm) (Y1); %drug encapsulation (Y2); and %drug release (Y3%), respectively. According to these levels, X1 concentrations at low, medium, and high were 0.1% *w*/*v*, 0.2% *w*/*v*, and 0.3% *w*/*v*; X2 concentrations at low, medium, and high were 0.2% *w*/*v*, 0.4% *w*/*v*, and 0.6% *w*/*v*; X3 concentrations at low, medium, and high were 16 mM, 24 mM, and 32 mM, respectively (Table 1). Two-dimensional counter plots delineated the impact of formulation variables on responses and 3D response-surface morphology, showing the impact of independent variables on responses, are shown in Figure 1 and Figure 2. The selection of ranges, low (−1) and high (+1), of independent variables was solely based on the preliminary examination of the independent variables, and was reported to be consistent and robust for development of the formulation. The quadratic model was found to be the best-fitted model with regard to coefficient of correlation (R^2^)~1 for investigating impacts of independent variables on responses. The polynomial equation generated based on this model established an individual, combined, and quadratic effect on responses. The BBD expressed 17 formulations with five center points to canvass any errors in the 5-times-replicated formulation, as shown in Table 2. Table 3 indicates the model summary statistics for regression analysis of responses Y1, Y2, and Y3 for data fitting data into various models viz. linear, 2FI, and quadratic model.

### 2.2. Impact of Independent Variables X1, X2, and X3 on Particle Size (Y1)

The equation expressing the impact of independent variables, on Y1, particle size is as follows;
Y1 = +142.00 + 4.63 × X1 + 3.25 × X2 + 9.38 × X3 + 15.75 × X1 × X2 − 1.00 × X1 × X3 − 10.75 × X2 × X3 − 12.50 × X1^2^ + 13.25 × X2^2^ + 8.40 × X3^2^

The impact of independent variables as suggested by quadratic equation and contour plot (Figure 1) and 3D response-surface morphology (Figure 2) expressed a considerable positive impact on particle size. The size distribution of chitosan NPs were narrow and reproducible, ranging from 121 nm to 180 nm. The lowest particle size obtained 121 nm at 0.1% *w*/*v* concentration of chitosan, and by increasing their concentration 0.3% *w*/*v*, the size of particles increased to 166 nm, which may be attributed to the increased viscosity. The high viscosity of the formulation may lead to binding to the alginate-gel matrix, resulting in increased particle size [16,17,18]. On the other hand, sodium alginate and chitosan both havepositive impacts on particle size. Increasing their concentration would lead to increased particle size due to complex formations with negatively charged alginate with cationic polymer chitosan, probably to due to the combined effect. Sodium alginate primarily forms complexes with calcium ions in the solution, and thereby forms a calcium-alginate pre-gel state, which is essential for interacting with cationic polymer chitosan. Calcium ions as a cross-linker further strengthen and stabilize the nanocarrier [19]. Many studies pointed out that increasing Ca2+ ion concentration in the formulation led to decreased particle size of chitosan/alginate NPs and improved drug entrapment [20,21,22]. The specific weight ratio of sodium alginate and chitosan had a significant impact on the fate of the in vitro performance of the NPs [16].

### 2.3. Impact of Independent Variables X1, X2, and X3 on %Drug Encapsulation (Y2)

The quadratic equation expressing the impact of independent variables on Y2 %drug encapsulation of chitosan nanoparticles isas follows:Y2 = +72.80 − 5.12 × X1 + 1.75 × X2 – 3.38 × X3 − 1.50 × X1 × X2 + 6.25 × X1 × X3 − 3.00 × X2 × X3 + 3.47 × X1^2^ +8.22 × X2^2^ – 1.52 × X3^2^

The impact of independent variables on %drug encapsulation as per the quadratic equation, 3D response-surface morphology (Figure 2) and contour plot (Figure 1) expressed a mixed impact on the %drug encapsulation. The encapsulation efficiency of chitosan NPs was found in the range of 67 to 93%. The increasing chitosan concentration showed a slight decrease in %drug encapsulation. Increasing chitosan concentration may increase the viscosity of the formulation, which may resist the entrapment of the drug within polymeric matrix. Adversely, the increased alginate concentration had a positive impact on encapsulation due to the gel-forming tendency of sodium alginate, which is capable of holding high drug concentrations. Bi et al. developed a hydroxyapatite-loaded microsphere composite of sodium alginate/chitosan, applying the emulsion-crosslinking method. The authors reported a high doxorubicin entrapment of 93.72% and loading of 46.86%, respectively, which demonstrated a higher drug-entrapment and retention capability due to the 3D-network-like structure [23]. The individual impact of the crosslinker Ca^2+^ lightly reduced the encapsulation efficiency of the chitosan NPs, which may due to the high porosity, swellability, and hydrophilicity of sodium alginate in the aqueous medium [24]. However, the combined impact of the crosslinker Ca^2+^ ion with oppositely charged polyelectrolyte chitosan led to improved encapsulation of the drug due to the protection or imparting of the complex structure of alginate/chitosan NPs [25]. Adversely, the combined impact of chitosan and alginate reducedthe encapsulation efficiency of the drug. This could be attributed to the fact that the binding site of alginate moiety was competitively partitioned out by chitosan drug molecules [26].

### 2.4. Impact of Independent Variables X1, X2, and X3 on %Cumulative Drug Release (Y3)

The quadratic equation expressing the impact of independent variables on Y3 %cumulative drug release of chitosan nanoparticles is as follows;
Y3 = +66.80 − 5.62 × X1 + 2.50 × X2 + 2.63 × X3 − 4.75 × X1 × X2 + 4.00 × X1 × X3 + 3.75 × X2 × X3 + 19.35 × X1^2^ − 8.90 × X2^2^ + 2.35 × X3^2^

The %drug release from β-SIT-Alg/Ch-NPs-FA varies in the range of 37% to 77%. The high chitosan concentration in the formulation had a negative impact on %cumulative drug release. This may be due to increasing viscosity, and thus reduces drug diffusion from the NPs into an encompassing medium. The reason for the low drug release was that high chitosan concentration increased the viscosity, and thereby the chances of particle agglomeration relatively increased, leading to increased particle size in the formulation. The smaller the particle size, the faster the dissolution in the surrounding medium, because the NPs will easily diffuse and readily dissolve, resulting in faster drug release [18]. The sodium alginate and Ca^2+^ concentration exerted a positive effect on the drug release profile. This can be explained by the alginate-forming gel-network structure with Ca^2+^, which rendered more drug molecules and at same time a high porosity, gelling property, and hydrophilic behavior of alginate, leading to high drug release [27].

### 2.5. Checkpoint Analysis of Optimum Formulation

The optimized formulation was selected based on the numerical method by keeping the criteria of minimizing particle size and maximizing %drug encapsulation and %drug release. The desirability value of the optimized formulation was found equal to 1; explaining the developed method for formulation was robust and strong. The smaller the particle size, the larger the surface area, contributing to better solubility and dissolution of the drug, and hence overall improving the drug availability from the nanosystem in the biological medium [28]. The composition of optimized alginate/chitosan NPs was chitosan (0.1% *w*/*v*); sodium alginate (0.48% *w*/*v*); and calcium chloride (16.32 mM). The statistically developed predicted value for the following responses: particle size; %drug encapsulation; and %drug release, were accounted at 121 nm, 93%, and 72.21%, and their corresponding observed value of particle size, %drug encapsulation, and %cumulative drug release were 126 ± 8.70 nm, 91.06 ± 2.6%, and 71.50 ± 6.5%, respectively. The drug-loading efficiency of the optimized formulation was estimated at 6.0 ± 0.52%. The optimum composition of independent variables observed vs. predicted responses in the Box Behnken design of β-SIT-Alg/Ch-NPs-FA is shown in Table 4. The variation in the predicted vs. observed value was lower and was statistically insignificant (*p* < 0.05), as shown in Table 4. Despite the above investigated parameters, the surface charge of the NPs was also determined as +25 mV, which explains the optimized formulation was robust and stable. Furthermore, a low polydispersity index of the formulation, indicating a uniform, homogeneous, and unimodel distribution of particles in the formulation, further confirmed the stability of the nanoformulation [29].

### 2.6. Characterization of Nanosystem

#### 2.6.1. Nanoparticlecharacterization

The particle size of β-SIT-Alg/Ch-NPs-FA was observed to be 126 ± 8.70 nm and the particle population in their formulation was uniform, consistent, and unimodal. The polydispersity index of formulation was low 0.211 suggested homogeneity and a monodispersed nanoparticulate system. The surface charge on β-SIT-Alg/Ch-NPs-FA was in fact found to be +25 mV using a zeta potential measuring instrument. The surface charges of chitosan-layered NPs were positive, due to the positively charged functional group on the chitosan molecule. The positive charge due to chitosan provided the colloidal stability to the NPs and portrayed an important prediction in the biological system. The folate-functionalized sodium alginate/chitosan NPs-FA showed an insignificant increase in particle size, and PDI confirmed that the surface-functionalization process of β-SIT-Alg/Ch-NPs with folic acid would not alter the physicochemical properties of the NPs. The TEM image received, indicating agreement with the investigation, was made on a zeta sizer (Figure 3). The in vitro characterization of physicochemical features of β-SIT-Alg/Ch-NPs-FA as the particle size distribution curve and surface charge from Malvern zeta sizer are furnished in Appendix A.

#### 2.6.2. Thermal Behavior

The thermal analysis results of β-SIT-Alg/Ch-NPs-FA, folic acid, and plain β-SIT are shown in Figure 4A–C. The thermal behavior corresponding to the DSC diagram indicatesthe endothermic peak of pure β-SIT at a melting point of 137.89 °C, showing a crystalline structure. However, complexation with sodium alginate and chitosan, and the resulting interaction, leads to the reduced crystallinity of the drug in the formulation, which is evidently shown by lowering in the melting point. The folic acid showedthe first endothermic peak at a melting point of 180.44 °C due to the presence of glutamate moiety, and the second exothermic peak was observed at a melting point of 232.76 °C due to the presence of amide moieties. The crystalline structure of folic acid changed into an amorphous state at above 200 °C, as shown as in the folate-conjugated formulation, β-SIT-Alg/Ch-NPs-FA Figure 4A [30,31].

#### 2.6.3. FT-IR Spectral Analysis

The chemical stability and physicochemical interaction among excipients in the formulation were assessed using the FT-IR spectroscopic technique. The results of the study revealed that the drug molecules were chemically stable and well-entrapped inside the polymeric NPs. The IR spectra of plain β-SIT (A), folic acid (B), and β-SIT-Alg/Ch-NPs-FA (C) are shown in Figure 5. The drug-molecule peaks are also indicated in the formulation, β-SIT-Alg/Ch-NPs-FA, indicating excipients’ compatibility with β-SIT in the formulation with other excipients. Therefore, the IR spectra corroborated the chemical stability of excipients in the formulation. The findings are consistent with the previous studies [32,33].

#### 2.6.4. Drug Release and Kinetics

The drug release from β-SIT-Alg/Ch-NPs-FA at various time points is expressed in Figure 6A–C. The release behavior from the polymeric system suggested a dual-release pattern; in early phase of drug release, it seemed to be an abrupt or fast release in 3 h, and in the later part it was a slow release (sustained release) for longer period of time, 48 h. The sustained-release behavior of the drug from the polymeric system, β-SIT-Alg/Ch-NPs-FA was accounted to release upto37.65 ± 7.95% vs. 7.43 ± 4.75% by β-SIT suspension over a time period of 48 h in PBS pH 7.4. The low drug release from the drug suspension may be attributed to the hydrophobic nature, poor aqueous solubility, and low dissolution. Adversely, at pH 6.5, which resembles the early endosomal pH of the tumor site, the drug release from the chitosan NPs were 69.21 ± 9.12% vs. 10.24 ± 4.65% released of β-SIT from drug suspension. The highest drug release from chitosan NPs observed at the end of 48 h was 71.50 ± 6.5% in PBS at pH 5.5 (late endosome) compared to 12.43 ± 2.51% drug released from β-SIT-suspension. The differences in drug release at pH 5.5 and pH 6.5 were statistically nonsignificant (*p* < 0.05) but significantly higher at pH 5.5 and pH 6.5 compared to biological fluid, PBS, pH 7.4. Furthermore, %cumulative drug release from β-SIT-Alg/Ch-NPs-FA was significantly higher over drug suspension in either dissolution medium (*p* < 0.05). The reason for the higher drug release in acidic conditions could be the prompt dissolution of polymeric complex between β-SIT and Alg/Ch complex in the acidic medium. It is worth mentioning that the pH in the tumor microenvironment is fairly more acidic than biological fluid and organs, and there is a disappearance of electrostatic interaction in the acidic pH. Some reports also showcased that the swelling of chitosan polymer is elicited when the pH drops to acidic. The dissolution study reported herein is in concurrence with the preceding works in literature [32,33,34,35]. The current study explained that the swelling behavior of the chitosan polymer is favored due to the repulsive forces exiting between the cationic charge in the polymeric chain complexes with alginate and Ca^2+^ ion, which triggers a higher drug release in pH 5.5 compared to pH 6.5 and pH 7.4. Moreover, the drug-release mechanism from alginate/chitosan polymer matrix depends on the aqueous incursion in the polymeric matrix, causing hydration accompanied with gel swelling, drug diffusion, and dissolution in the surrounding medium. Alginate was found to exert an influence on the drug release, and the particles with the maximum alginate mass fraction showed sustained release through the dissolution mechanism. During this process, part of the gel-polymer matrix was broken or eroded into the medium. It is worthwhile to note that the drug release from the polymeric system depends on the pH of the medium, dose of drug, and physicochemical behavior of the nanocarrier system [36,37].

The drug releases of β-SIT-Alg/Ch-NPs-FA were fitted to various kinetic models to come across the fate of the drug release, viz. first-order, zero-order, Higuchi, and Peppas, Hixson–Crowell, and to select the most effective model based on the good data fit. The selection of the release kinetics model, in reply to the coefficient of correlation (R^2^) estimated from different models, stated that the best-fitted model for drug release from NPs was the Korsmeyer–Peppas release model, with the regression value recorded (R^2^ = 0.9632). Moreover, the estimated *n* exponent value was 0.420 (0.5 < *n* < 1), suggesting that the drug-release mechanism from β-SIT-Alg/Ch-NPs-FA complied with the Fickian diffusion [18,38]. The drug release from the NPs was generally controlled by the penetration of water in the polymeric matrix, followed by hydration, and then matrix swelling and erosion of the matrix gelatinous mass, and thereby diffusion of the dissolved drug into the medium. Despite this, the release rate could also be influenced by the nature of the drug, its dose, the pH of surrounding medium, and the nature of the polymer in which the drug is entrapped [18].

#### 2.6.5. Everted Gut Sac Model

The intestinal permeation study outcomes are clearly depicted in Figure 7, which reveals that an extremely significant quantity of β-SIT permeated and was transported through the intestinal mucosa from the optimized β-SIT-Alg/Ch-NPs-FA as compared to β-SIT-suspension. Furthermore, the effect of folate conjugation over the chitosan surface had no effect on permeation across the intestinal mucosa. The cationic-charge surface on the chitosan polymer has positive impact on their transport due to the substantial interaction with the negative-charge surface of the mucosal membrane, and helps in opening the intestinal tight junction. This may lead to a higher amount of β-SIT permeation and transport through the intestinal mucosal membrane [39].

#### 2.6.6. Cell-Viability Assay

The technique of evaluation of cell toxicity based on MTT assay relies on the mechanism of the reduction in yellow-color MTT dye by viable cells into formazan crystals (purple). The results of the reduction in cell viability post-formulation treatment were found to be concentration- and time-dependent. After a 24 h period of assessment, the percentage (%) of cell viability was reported to be 39% and 87%, corresponding to the formulation β-SIT-Alg/Ch-NPs-FA and β-SIT-suspension. At the end of 48 h of treatment, the %cell viability was determined to be 15.23% and 79% by β-SIT-Alg/Ch-NPs-FA and β-SIT-suspension, as shown in Figure 8. The concentration at which 50% of the cell death took place was the inhibitory concentration (IC_50_), at which the estimations for the formulation β-SIT-Alg/Ch-NPs-FA and β-SIT-suspension were 37.41 ± 5.30 and 189 ± 10.23 μM at the end of 24 h. Furthermore, the IC_50_ was subsequently reduced to 25.75 ± 1.56 μM and 123.50 ± 8.05 μM, accompanying the formulation β-SIT-Alg/Ch-NPs-FA and β-SIT-suspension after 48 h of incubation in the cell line. The outcomes of the study suggested that the cytotoxic effects caused by β-SIT-Alg/Ch-NPs-FA to breast carcinoma cells were extremely significant (*p* < 0.001) when compared to β-SIT-suspension.

Notably, nanocarriers having anticancer agents exhibit antiproliferative effects, primarily on the passive-targeting or EPR-based approach, which enables the transportation ofthe drug from the nanocarrier onto the cells. The mechanism of an EPR-based drug-targeting relies on the availability of drug concentration and permeation of the therapeutics at the target site or nearby to the cancerous cells/tissues, as well as in the microenvironment of the tumor type [40,41]. The polymer particles protect the phytoactive from degradation, providing stability and releasing content transported to the cells mediated via passive targeting in the physiological medium, and interacting with biological components such as the protein fraction of blood, and thereby shows an enhanced antiproliferation effect against the cancer cells [42,43]. On the other hand, the actively targeted polymer particles (β-SIT-Alg/Ch-NPs-FA) bind with the receptor onto the cell surface, thereby releasing therapeutics specifically into the tumor endosomes and nucleus, resulting in enhanced death of tumor cells and causing less or no harm to the neighboring cells. The higher antiproliferation effect shown by β-SIT from such nanocargo was probably due to higher solubility, dissolution, and inherent targeting capabilities. Adversely, the lesser cell death, reported from the β-SIT-suspension due to their unformulated and hydrophobic nature, limited their activity [44,45].

#### 2.6.7. Antioxidant Activity

The antioxidant assay used DPPH, a stable free radical widely used for the measurement of antioxidant potential in a number of phytoconstituent and drug substances [46]. The DPPH as a free radical accepts protons from the donor group for which antioxidant capacity is to be measured. It was found that the antioxidant activity depended on the concentration of β-SIT in the different samples. The comparative antioxidant properties of β-SIT with respect to β-SIT-Alg/Ch-NPs-FA are shown in Figure 9. The antioxidant capacity of the β-SIT-suspension and β-SIT-Alg/Ch-NPs-FA were determined to be 91 ± 5.99%, and 29 ± 8.02%, respectively. Moreover, the results revealed that %inhibition by folate-functionalized prepared β-SIT-Alg/Ch-NPs-FA was much more significant than the β-SIT-suspension (*p* < 0.001). The higher antioxidant property of β-SIT-Alg/Ch-NPs-FA was due to the availability of the encapsulated drug in a highly dissolved state interior to the polymeric core, which gave protection and stability against the external environment from degradation.

#### 2.6.8. Stability Study

The stability study as per the guidelines was successfully performed. The regular investigation of samples at various intervals concluded that alterations in the particle size (nm), PDI, surface charge, %drug entrapment, and %drug retention on the NPs were less prominent, further clarifying thatthe developed formulation was robust and stable within the specified time. Although the level of changes in these parameters post-assessment was recorded to be relatively higher in the formulation stored at an elevated temperature, changes were statistically nonsignificant (*p* > 0.05). The stability profile of the formulation, β-SIT-Alg/Ch-NPs-FA under different conditions is shown in Figure 10.

## 3. Conclusions

The folate-functionalized β-SIT-Alg/Ch-NPs-FA were successfully designed and developed for improved drug delivery and efficacy against breast cancer cells. The impact of the chosen variables—chitosan (X1), sodium alginate (X2), and calcium chloride (X3)—concentrated with three different levels—low (−1), medium (0), and high (+1)—were satisfactorily studied on the responses of particle size (nm) (Y1); %drug encapsulation (Y2); and %drug release (Y3). The optimized formulation, ascertained by the composition X1:X2:X3, was 0.1%:0.48%:16.32 mM, reporting a robust and stable formulation. The electron microscopic image of NPs showed a spherical shape; particles were consistent and uniformly distributed, which further corroborated the particle size finding by Malvern Zetasizer. The Alg/Ch polymer modulated the β-SIT release from NPs and established a sustained release profile of 71.50 ± 2.45% drug release in PBS at a late endosomal pH for a period of 48 h. The best-fitted model for the drug release from NPs was the Korsmeyer–Peppas release model, with the regression value recorded (R^2^ = 0.9632) and the release mechanism complying with the Fickian mode of drug diffusion. Furthermore, the analytical studies of developed formulation, viz. DSC and FT-IR, corroborated the physicochemical stability with reduced crystallinity. The cell-viability study revealed that phytosterol from folate-conjugated NPs was more effective than β-SIT-suspension in the breast cancer line. The β-SIT-Alg/Ch-NPs-FA expressed higher intestinal permeability than the β-SIT-suspension. Furthermore, the β-SIT-Alg/Ch-NPs-FA demonstrated a higher inhibition of free radicals compared to the β-SIT-suspension. Considering all the outcomes of the present study, it suggests that the developed formulation is promisingly useful in the treatment of breast cancer.

## 4. Materials and Methods

### 4.1. Materials

β-Sitosterol (β-SIT), Chitosan (MW = 50–190 KDa), sodium alginate, and folic acid were received from Sisco Research Laboratories Pvt. Ltd. EDC and sulfo-NHS reagent obtained from Sigma-Aldrich; calcium chloride was received from Qualigens fine chemicals (Mumbai India); ethanol was obtained from Merck Pvt. Ltd. (Mumbai, India). Analytical-grade reagents such as HPLC water, solvents, and other chemicals were used in the analysis. The cells were obtained from the National Centre for Cell Science (NCCS) (Pune, India). The cell-viability experiment was conducted on breast-cell lines, MCF-7, as per standard procedure mentioned in the literature. The culture medium for growing cells wasstreptomycin (100 mg/mL) withDulbecco’s modified Eagle medium (DMEM); Fetal Bovine Serum (FBS) (10%); and penicillin (100 unitmL^−1^) in well-defined conditions. The cells were monitored regularly and kept under incubation at 37 °C and 5% CO_2_/95% air environment. The buffering agent, phosphate saline buffer (PBS), was used foranalytical grades, and their components, viz. sodium hydroxide, sodium dihydrogen phosphate, potassium dihydrogen phosphate, disodium hydrogen phosphate, and organic solvent (glacial acetic acid, acetonitrile, ethanol) were obtained from the central drug house (New Delhi, India).

### 4.2. Analytical Methodology

High-performance liquid chromatography (HPLC) was used for the separation of β-SIT. The components of HPLC system were a binary pump (model 1525; Milford, CT, USA); and a C_18_ column with configuration of (150 mm × 3.9 mm × 5 μm). The analyte was separated out using a mixture of mobile phase of acetonitrile: methanol 70:30 (*v*/*v*) at a flow rate of 1 mL. The injection volume of sample was 10 μL. A durapore membrane filter of size 0.21 µ was used with injectable syringe for sampling of analyte. Amembrane filter of pore size 0.45 µ was used for filtering the entire sample used under analytical study.

### 4.3. Formulation Optimization

The time-effective, low-cost and economic statistical 3-factor, 3-level Box–Behnken design was used to optimize the Alg/Ch-NPs. Design-Expert software (Version 10; Stat Ease Inc., Minneapolis, MN, USA) [28,47,48] was employed in the optimization process. The levels of independent variables used in this study are expressed in Table 1. The impact of alteration of independent variables, viz. chitosan (X1), sodium alginate (X2), and calcium chloride (X3), were investigated in the responses, viz. particle size, (Y1), %drug encapsulation, (Y2) and %drug release (Y3), for developing NPs, as shown in Table 2. The best-fitting model was analyzed and adopted according to ANOVA for optimizing the formulation. The point-prediction method was employed, keeping criteria of minimum Y1, and maximum Y2 and Y3, to finalize the optimum formulation of β-Sitosterol-loaded Alg/Ch-NPs. The statistical analysis was used to assess the independent variables affecting the responses, as well as the interaction between the factors. The multivariate linear regression equation, accounting the significant relationship among the independent variables and responses, is as follows:Y = λo + A_1_Z_1_ + A_2_Z_2_ + A_3_Z_3_ + A_12_Z_1_Z_2_ + A_13_Z_1_Z_3_ + A_23_Z_2_Z_3_ + A_11_Z_1_^2^ + A_22_Z_2_^2^ + A_33_Z_3_^2^
where Y indicates the measured responses or dependent variables; λo is the intercept; A_1_ to A_33_ are regression coefficients of different variables; while Z_1_, Z_2_, and Z_3_ are the coded levels of independent variables. Z_1_Z_2_, Z_1×3_, Z_2_Z_3_, and Z_l_^2^ (l = 1, 2, 3) show the interaction and quadratic effects. The levels followed in the experimental base design were low, medium, high, axis, and central point. The statistical analysis revealed the best-fitting model among the different models, viz. linear, quadratic, cubic, and 2FI, was quadratic and selected it for the study. The best-fitting model was indicated by F-value, low PRESS, and insignificant lack of fit in the proposed model.

### 4.4. Preparation of β-Sitosterol-Loaded Alg/Ch-NPs

The β-Sitosterol-loaded Alg/Ch-NPs were prepared as per previous work with modification [15,49]. The β-SIT-Alg/Ch-NPs were prepared by cation-induced gelation technique or ionotropic gelation technique. Primarily, 5 mL of sodium-alginate (0.48% *w*/*v*) solution was prepared by dissolving in deionized water and adjusting the pH (with 1 M HCl) to 5.2. β-Sitosterol (5 mg) was dissolved in ethanol and transferred dropwise into alginate solution until a homogeneous preparation was formed. Then, calcium chloride solution (5 mL) was added to alginate solution dropwise (0.5 mL/min) with continuous magnetic stirring at 1000 rpm for 30 min, resulting in a solution forming a complex of calcium-drug-alginate beads. The chitosan was dissolved in glacial acetic acid of 1% (*v*/*v*) to form a solution. Then, 10 mL of chitosan solution was incorporated dropwise into the alginate solution, leading to formation of self-assembled NPs. Moreover, Alg/Ch-NPs were magnetically stirred for 1 h to reach homogeneity, and disperse particles were thereafter sonicated (Probe sonicator, Hielscher Ultrasonics, Berlin, Germany) for 10 min (one cycle, 30 kHz power, 80 W), and centrifuged (OptimaTM LE-80K Ultracentrifuge) 15 min at 15,000× *g*; and post-equilibration, NPs were washed, filtered, and lyophilized for future use. The sketch expresses step I, which involves the formulation of β-SIT-Alg/Ch-NPs, and step II, which indicates folate modification of NPs, as shown in Figure 11.

### 4.5. Folate Modification of β-Sitosterol-Loaded Alg/Ch-NPs

β-SIT-Alg/Ch-NPs (10 mg) were dissolved in (PBS, 1 mL). The EDC (20 mg/mL) solution in deionized water was injected slowly into above solution with continuous magnetic stirring. After that, the sulfo-NHS was prepared in solution (sulfo-NHS dissolved in PBS, pH 7.4) then added into it with uninterrupted magnetic stirring. The resulting β-SIT-Alg/Ch-NPs solution was then placed on a biological shaker overnight for surface activation. On the next day, the NPs were centrifuged 15,000× *g*, the supernatant was discarded, and residual NPs washed with PBS. Moreover, dried NPs were redispersed in 1 mL buffer solution of pH 7.4, and FA in 5 mL NaOH (8.828 mg/mL) was dissolved into them, and was continuously shaken in biological shaker under dark conditions for 24 h for surface conjugation to NPs. The suspension of folate-conjugated NPs was centrifuged at 10,000× *g* for 30 min, and NPs were washed in PBS, separated from unentrapped folate, dried, and lyophilized for further use. The FA-conjugated NPs were designated to amino groups of chitosan to FA-leveled β-SIT-Alg/Ch-NPs-FA.

### 4.6. In Vitro Characterization of Nanosize System

#### 4.6.1. Particle Sizes and Their Distribution

The particle size and their distribution of β-SIT-Alg/Ch-NPs-FA were analyzed by applying Zetasizer Nano ZSP (Malvern Instruments, Worcestershire, UK). The formulation was 10-fold diluted in deionized water, and afterwards, sonication particle size was measured. The studies were performed in triplicate (*n* = 3). The surface charge on the β-SIT-Alg/Ch-NPs-FA was also measured as zeta potential. The particle size was further evaluated using TEM instrument (Techni TEM 200 Kv, Fei, Electron optics). The 20 μL of diluted sample was spread over copper grid (carbon coated); post-drying process, it was coated with 1% phosphotungstic acid and observed under microscope at accelerating voltage of 100 kV. The images of the nanodispersed system were captured and assessed, applying image-analyzing software.

#### 4.6.2. Drug Encapsulation and Loading

The quantity of drugs entrapped in the β-SIT-Alg/Ch-NPs-FA was estimated before lyophilization. The NPs were centrifuged at high speed 10,000× *g* for 30 min (REMI Cooling Centrifuge, India). The amount of free drug in the supernatant was separated from β-SIT-Alg/Ch-NPs-FA dispersion and unentrapped, or the free drug in the supernatant further diluted and quantified using HPLC. The %drug encapsulation efficiency and %drug loading were estimated using the formula:EE (%) = Total quantity of β-SIT/Total quantity of β-SIT in supernatant/Total quantity of β-SIT × 100
DL (%) = β-SIT encapsulated in β-SIT-Alg/Ch-NPs-FA/Total quantity of β-SIT-Alg/Ch-NPs-FA weight × 100

#### 4.6.3. Fourier Transform Infrared Spectroscopy (FT-IR)

The β-SIT, folic acid, and surface-functionalized formulation β-SIT-Alg/Ch-NPs-FA were analyzed using FT-IR spectrometer (BRUKER Corporation, Billerica, MA, USA). The spectrum of the different samples was recorded and analyzed. The measured quantities of pure drug and β-SIT-Alg/Ch-NPs-FA ~5 mg were placed in direct contact with a light beam, and spectrum generated was recorded in the scanning range of 3500–400 cm^−1^.

#### 4.6.4. Differential Scanning Calorimetry

The enthalpy change (∆H) of pure β-SIT, folic acid, and β-SIT-Alg/Ch-NPs-FA were analyzed by differential scanning calorimetry (Pyris 4 DSC, Perkin Elmer, Waltham, MA, USA). The sample of approximately ~5 mg and reference standard were both placed on different DSC aluminum pans. The heating started on both the pan simultaneously at a scanning rate of 10 °C/min (20–350 °C) with the use of effluent gas as dry nitrogen.

#### 4.6.5. Drug-Release Study

The drug release from β-SIT-Alg/Ch-NPs-FA was investigated in vitro in simulated intestinal fluid (SIF), viz. PBS of pH 7.4, and compared with β-SIT-suspension. The study was acquitted using dialysis membrane, which was preactivated in PBS solution before the offset of the release study. The measured quantity (10 mg) of β-SIT-Alg/Ch-NPs-FA, and β-SIT-suspension carrying the same drug dose were enfolded into a dialysis bag having 95 mL of PBS (MW; 8–12 kDa; Repligen, Waltham, MA, USA) with clipped ends, and inserted into a release medium preheated with 37 ± 0.5 °C with continuous stirring [29]. The samples (1 mL) were withdrawn at a set point of time (0, 8, 16, 24, 32, 40, and 48 h), and same volume was supplanted with fresh dissolution medium. The picked-up samples were filtered, diluted, and quantified for drug content using HPLC. Furthermore, the drug-release analysis was fitted into various kinetic-release models using the graphical method to canvass the drug-release mechanism from NPs, and the model with good data fit was predicted.

#### 4.6.6. In Vitro Permeation Study

To investigate the intestinal permeability, the intestinal part from rat was isolated toperform the in vitro permeation study. Intestine was excised from Sprague Dawley rats weighing 180–200 g. All animals were handled in compliance with regulations of the law of ethics of research on living creatures in the Kingdom of Saudi Arabia. The study was approved by the Research Ethics Committee at Faculty of Pharmacy, King Abdulaziz University, Jeddah, Saudi Arabia (Reference No. Ph_1443_07). The cell debris, mucous, and foreign particles were properly removed with thorough washing with PBS of pH 6.8. The amount of β-SIT permeated across the rat intestinal mucosa at different time intervals from β-SIT-suspension, and β-SIT-Alg/Ch-NPs-FA has been shown in Figure 7. The formulation containing 5 mg of β-SIT was carefully transferred into the intestinal sac with each terminal end of the organ tightly ligated. Thereafter, the sample was incorporated in a receiver compartment containing 100 mL of PBS and aerated continuously, and the temperature of the compartment was kept constant at 37 ± 0.5 °C, along with constant magnetic stirring at 100 rpm. At predefined time period, 2 mL of sample volume was withdrawn and supplanted with same volume offresh medium. The drug content in the sample was quantified, applying HPLC method at 210 nm. The amount of drug permeation and permeation flux was determined [50,51].

#### 4.6.7. Cell-Viability Study

In cell viability study, the different concentrations of β-SIT-Alg/Ch-NPs-FA, and pure drug suspension withsimilar dose of drug were incubated in breast cancer cell line. Approximately 5 × 10^3^ cells/well weretransferred into a 96-well plate and incubated at 37 °C for 24 h. Then, cells were treated with different concentrations of β-SIT varying from 10–50 μM in β-SIT-Alg/Ch-NPs-FA and pure drug suspension with equivalent drug dose, and incubated for 24 h and 48 h at 37 °C. Thereafter, medium was replaced with MTT reagent (0.5%, ~10 µL) to eachwell and incubated for 4 h. Furthermore, the supernatant was removed and 100 µL of dimethyl sulfoxide (DMSO) was added to each well. The optical density was measured and recorded on a microplate reader (Bio-Rad, USA) at λ_max_ of 570 nm [52]. Untreated cells or blank formulations were used as control (100% viability). The IC_50_, the drug concentration in formulation which killed 50% of viable cells compared to control cells, was also determined. The cell viability (%) was assessed as average viability (%) ± standard deviation (SD) (*n* = 3) and calculated using equation:% Cell viability = Optical density of treated sample/Optical density of controlled × 100

#### 4.6.8. Radical Scavenging Power

The comparative antioxidant activity of β-SIT in optimized β-SIT-Alg/Ch-NPs-FA and β-SIT-suspension was assessed using 2, 2-diphenyl-2-picrylhydrazyl (DPPH) assay as per reported method with modification [53]. The stock solution of concentration (10 mg/mL) of the formulation β-SIT-Alg/Ch-NPs-FA and β-SIT-suspension was prepared in ethanol. From stock solution, the working standard of formulation and drug suspension were prepared in the range of (10–100 μg/mL). From each sample, concentration volume of 0.5 mL was mixed with freshly prepared ethanolic solution of DPPH reagent (0.02%, 100 µL) at ambient temperature with vigorous shaking and incubated for 30 min at 28 °C in dark. After a chemical reaction between antioxidant molecule and reagent, the violet-color solution changed to colorless. The blank mixture was prepared using 50 μL of ethanol. Later on, the decreased optical density of the formulation was measured at 517 nm with UV-Vis spectroscopy. The antioxidant power of drug in optimized β-SIT-Alg/Ch-NPs-FA, and β-SIT-suspension were measured as %inhibition or radical scavenging action in DPPH assay. The decreased DPPH concentration was estimated from the calibration plots of the samples, and the assay was repeated in triplicate (*n* = 3).

The % antioxidant activity was evaluated by applying equation:%Antioxidant activity = Ao − A1/Ao × 100,
where Ao is the blank; A1 is the sample absorbance calculated using UV-Visspectroscopy.

IC_50_ (μg/mL) was determined using dose–response curve of linear regression graph, and concentration achieved enabled 50% inhibition in free radical by DPPH radical.

### 4.7. Stability Study

The colloidal stability of developed optimized formulation β-SIT-Alg/Ch-NPs-FA was carried out as per the International Conference on Harmonization (ICH) guidelines for three months. The stability was measured on PBS of pH 7.4. For the study, β-SIT-Alg/Ch-NPs-FA was placed in a stability chamber at a temperature of 5 ± 3 °C, at ambient temperature 30 ± 2 °C/65 ± 5% RH), and at elevated temperature of 40 ± 2 °C/75 ± 5% RH [54]. The β-SIT-Alg/Ch-NPs-FA was monitored at intervals of 0, 1, 2, and 3 months by withdrawing and analyzing samples using HPLC for alteration in particle size, surface charge, PDI, %drug entrapment, and %drug retained. The study was conducted in triplicate (*n* = 3) [55].

### 4.8. Statistical Analysis

The sample analysis was executed statistically, applying one-way ANOVA followed by Tukey–Kramer analysis using GraphPad prism (version 7). The data obtained were explicated as mean ± SD for three samples (*n* = 3). The level of statistical significance (*p* < 0.05).

## Figures and Tables

**Figure 1 gels-08-00219-f001:**
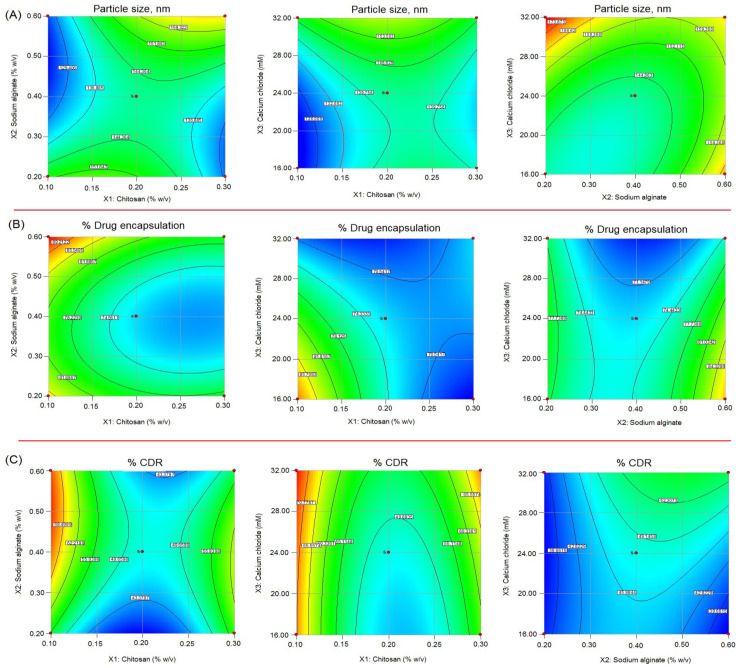
2D contour response-surface plots showing the influence of chitosan (X1), sodium alginate (X2), and calcium chloride (X3) on response parameters: (**A**) particle size (nm); (**B**) %drug encapsulation and; (**C**) %cumulative drug release (%CDR) of β-SIT-Alg/Ch-NPs-FA.

**Figure 2 gels-08-00219-f002:**
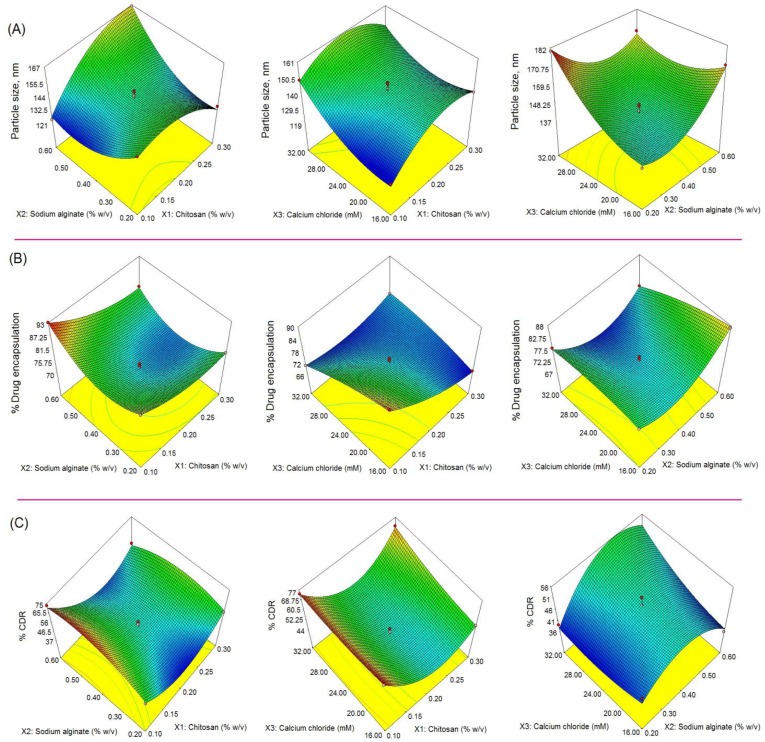
The response-surface curve (3D) (**A**–**C**) showing the effect of formulation variables: X1: Chitosan *(% w*/*v*), X2: Sodium alginate *(% w*/*v*), and X3: Calcium chloride (mM) on responses, particle size, nm (**A**), %drug encapsulation (**B**), and %cumulative drug release (%CDR) (**C**) of β-SIT-Alg/Ch-NPs-FA.

**Figure 3 gels-08-00219-f003:**
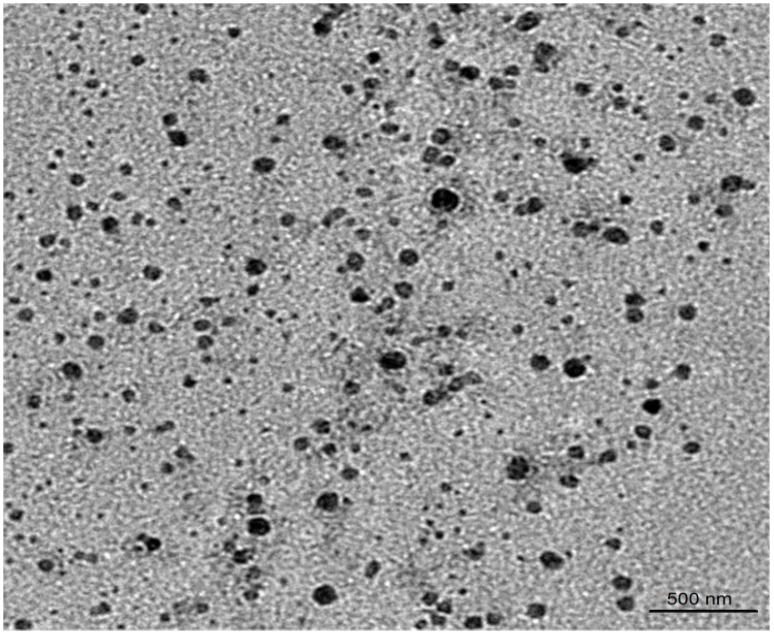
TEM image of β-SIT-Alg/Ch-NPs-FA.

**Figure 4 gels-08-00219-f004:**
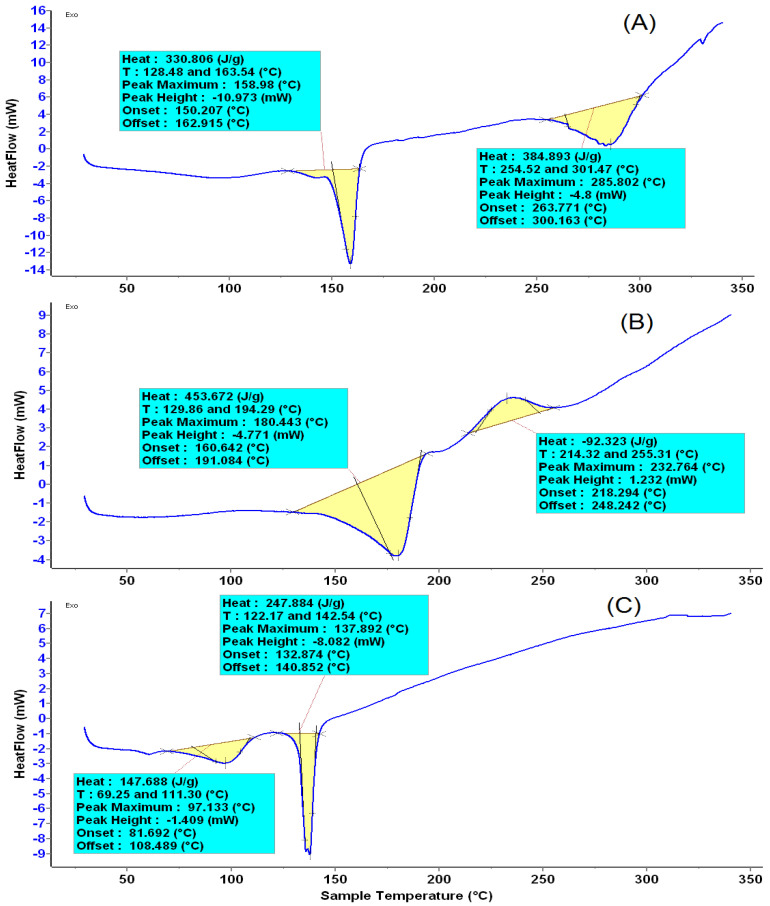
The melting-point thermogram of β-SIT-Alg/Ch-NPs-FA (**A**); Folic acid (**B**); and β-SIT (**C**).

**Figure 5 gels-08-00219-f005:**
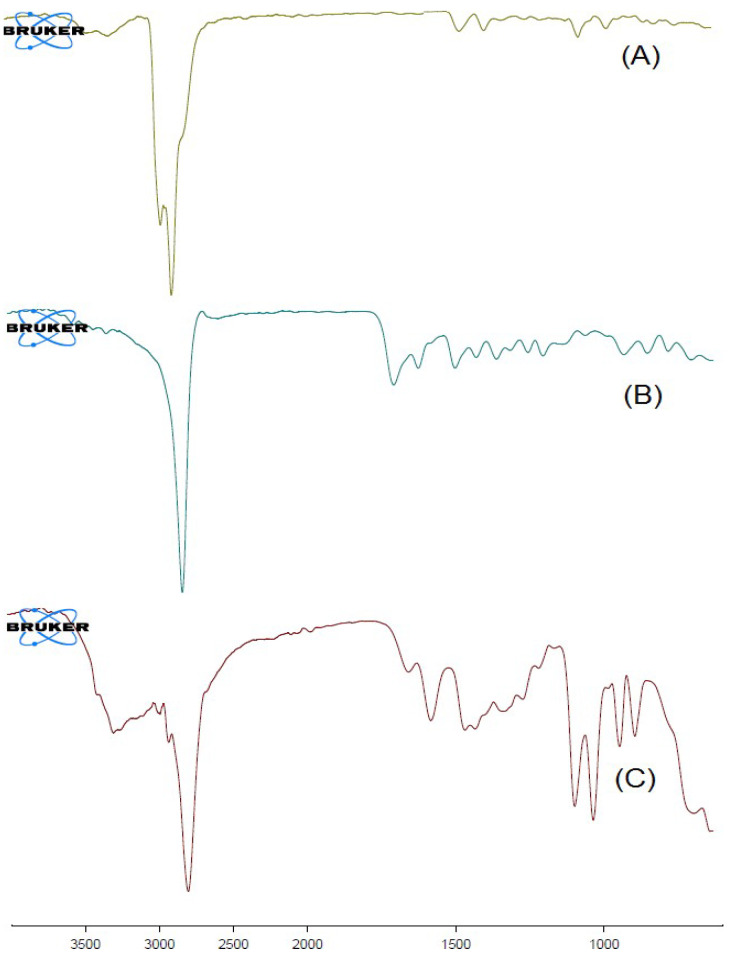
FT-IR spectra of β-SIT (**A**); folic acid (**B**); and β-SIT-Alg/Ch-NPs-FA (**C**).

**Figure 6 gels-08-00219-f006:**
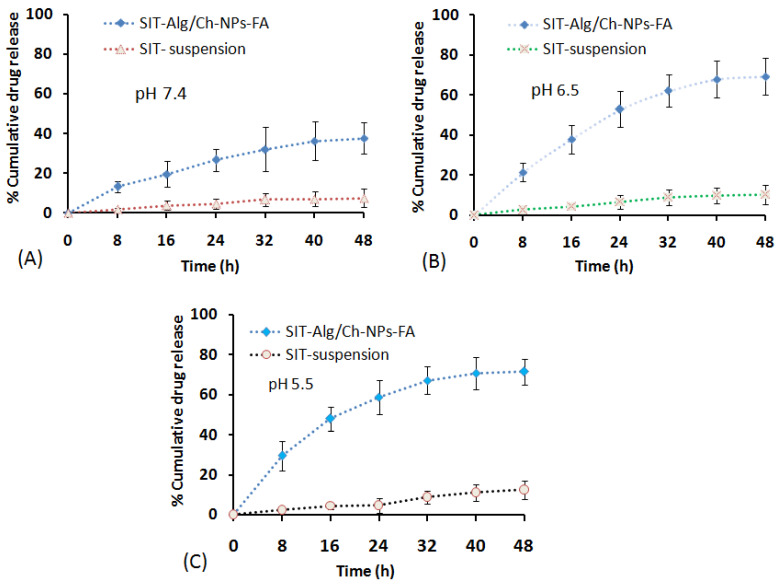
%Cumulative β-SIT release from SIT-Alg/Ch-NPs-FA and β-SIT-suspension in dissolution medium of different pHs: 7.4 (**A**); pH 6.5 (**B**); and pH 5.5 (**C**) at various time intervals of 0, 8, 16, 24, 32, 40, and 48 h, respectively.

**Figure 7 gels-08-00219-f007:**
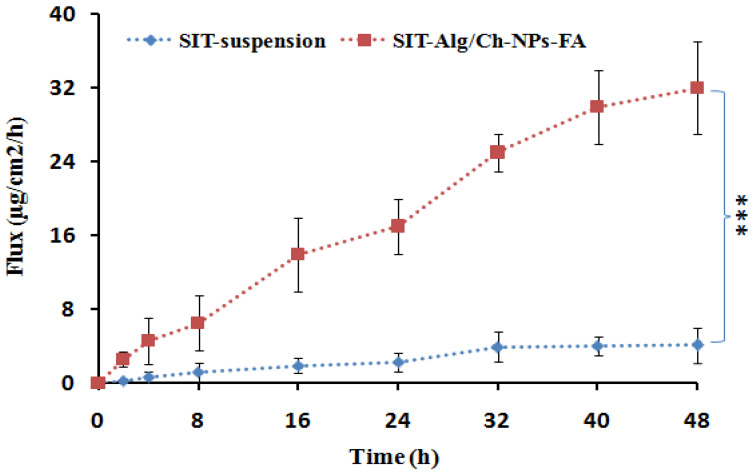
The flux of β-SIT-Alg/Ch-NPs-FA in comparison to β-SIT-suspension in ex vivo intestinal permeation study. Data expressed as mean ± SD (*n* = 3) (*** *p ≤* 0.001).

**Figure 8 gels-08-00219-f008:**
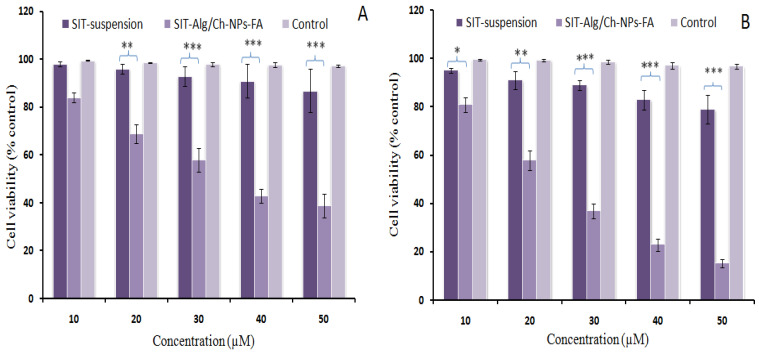
Percentage cell-viability assessment of β-SIT-Alg/Ch-NPs-FA and SIT-suspension at varying concentrations of (10, 20, 30, 40, and 50 µM) after incubation time of 24 h (**A**) and 48 h (**B**) in breast cancer cell line. Data expressed as mean ± SD (*n* = 3). Significant levels were (* *p* < 0.05), (** *p* < 0.01), (*** *p* < 0.001) when formulation SIT-Alg/Ch-NPs-FA compared with SIT suspension.

**Figure 9 gels-08-00219-f009:**
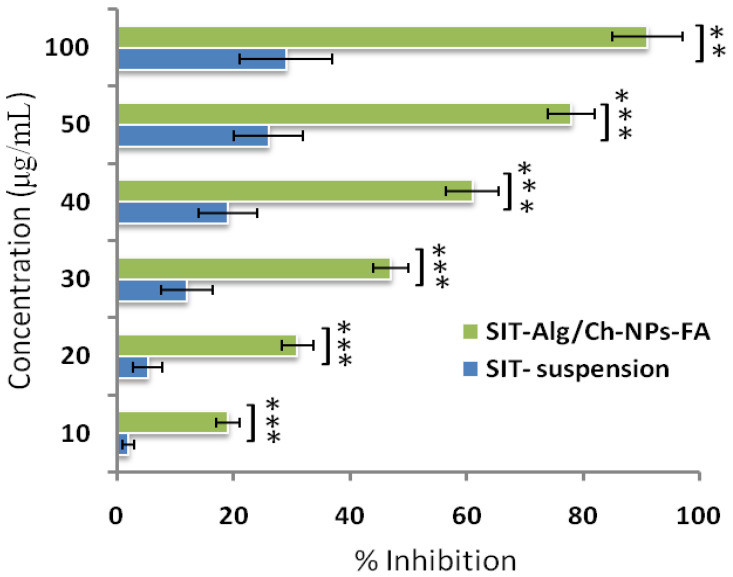
The antioxidant activity by DPPH assay of Alg/Ch-NPs-FA in comparison with β-SIT-suspension. Data expressed as mean ± SD (*n* =3). Level of significance (** *p* < 0.01, *** *p* < 0.001), highly significant when compared with β-SITsuspension.

**Figure 10 gels-08-00219-f010:**
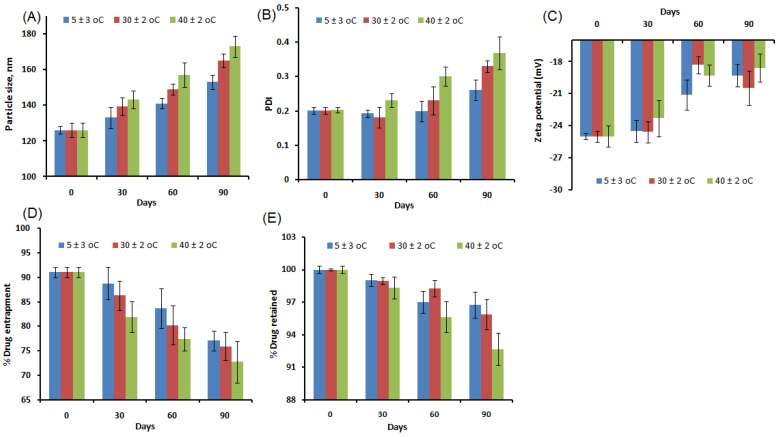
Stability indicating assessment of β-SIT-Alg/Ch-NPs-FA under different temperature conditions (5 ± 3 °C, 30 ± 2 °C, and 40 ± 2 °C) showing the changes in the particle size (**A**); PDI (**B**); zeta potential (**C**); %drug entrapment (**D**); and %drug retained (**E**) during 3 months. The study demonstrated that no significant alterations were reported in these parameters under study (*p* > 0.05) at the specified condition.

**Figure 11 gels-08-00219-f011:**
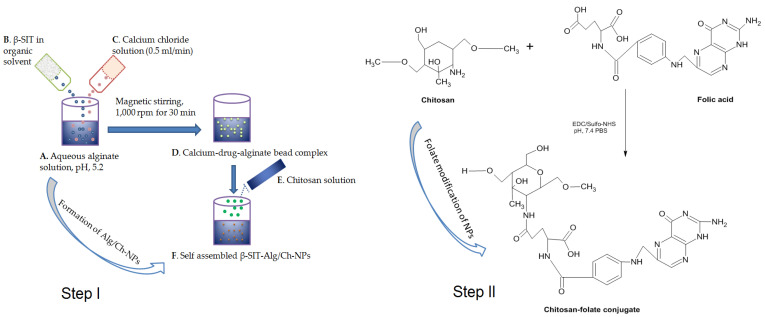
Outline representing the different steps in the preparation of β-SIT-Alg/Ch-NPs. Aqueous alginate (1 mg/mL) solution pH adjusted to 5.2 (**A**); Transfer of β-SIT solution in alginate solution (**B**); Calcium-chloride solution added to alginate solution (**C**) at flow rate of (0.5 mL/min) with magnetic stirring, 1000 rpm for 30 min, forming a complex of calcium-drug-alginate beads (**D**); Chitosan solution added dropwise to alginate solution (**E**), leading to formation of self-assembled β-SIT-Alg/Ch-NPs (**F**). Step I involves formulation of β-SIT-Alg/Ch-NPs, while Step II indicatesfolate modification of NPs.

**Table 1 gels-08-00219-t001:** The independent variables and levels employed as low (0), medium (−1), and high (+1) in statistical optimization of β-SIT-Alg/Ch-NPs-FA.

Independent Variables	Level Employed		
	Low (−1)	Medium (0)	High (+1)
X1: Chitosan (% *w*/*v*)	0.1	0.2	0.3
X2: Sodium alginate (% *w*/*v*)	0.2	0.4	0.6
X3: Calcium chloride (mM)	16	24	32
**Dependent Variables**			
Y1: Particle size (nm)		Minimize	
Y2: Drug encapsulation (%)		Maximize	
Y3: Drug release (%)		Maximize	

**Table 2 gels-08-00219-t002:** The in vitro investigated results of different formulations (FN1-FN17) for responses Y1, Y2, and Y3 in Box Behnken design for optimization of β-Sitosterol-loaded Alg/Ch-NPs-FA.

Formulation Code	Independent Variables			Responses		
	X1 (*% w*/*v*)	X2 (*% w*/*v*)	X3 (mM)	Y1 (nm)	Y2 (%)	Y3 (%)
FN1	0.10	0.40	16.00	121	90	70
*FN2	0.20	0.40	24.00	142	72	49
FN3	0.30	0.60	24.00	166	80	54
FN4	0.10	0.20	24.00	151	86	55
*FN5	0.20	0.40	24.00	137	73	47
FN6	0.20	0.60	16.00	170	87	37
*FN7	0.20	0.40	24.00	145	74	49
FN8	0.10	0.60	24.00	123	93	74
FN9	0.30	0.40	32.00	146	72	72
FN10	0.20	0.60	32.00	168	75	52
FN11	0.20	0.20	16.00	139	78	40
*FN12	0.20	0.40	24.00	140	74	48
FN13	0.10	0.40	32.00	149	70	77
FN14	0.20	0.20	32.00	180	78	38
*FN15	0.20	0.40	24.00	146	71	47
FN16	0.30	0.40	16.00	138	67	58
FN17	0.30	0.20	24.00	131	79	58

X1: Chitosan (*% w*/*v*); X2: Sodium alginate (% *w*/*v*); X3: Calcium chloride (mM); Y1: Particle size (nm); Y2: %Drug encapsulation; Y3: %Cumulative drug release.*Replicas.

**Table 3 gels-08-00219-t003:** Model summary statistics for regression analysis of responses Y1, Y2, and Y3 for data fitting data into various models.

Model	R2	Adjusted R^2^	Predicted R^2^	SD	CV %	Desirability
**Response: Y1**						0.958
Quadratic	0.9805	0.9554	0.8706	3.45	2.35	
2FI	0.5886	0.3417	−0.9643	13.25		
Linear	0.2245	0.0456	−0.5873	15.96		
**Response: Y2**						0.958
Quadratic	0.9904	0.9780	0.9565	1.11	1.42	
2FI	0.5934	0.3494	−0.5409	6.01		
Linear	0.3668	0.2207	−0.1936	6.58		
**Response: Y3**						0.958
Quadratic	0.9943	0.9870	0.9326	1.41	3.46	
2FI	0.23391	−0.2174	−2.5687	13.70		
Linear	0.1440	−0.0536	−0.7143	12.69		

**Table 4 gels-08-00219-t004:** Optimum composition of independent variables, observed vs. predicted responses in BBD of β-SIT-Alg/Ch-NPs.

Independent Variables	Optimized Composition	Observed Responses			Predicted Response		
		Y1 (nm)	Y2 (%)	Y3 (%)	Y1 (nm)	Y2 (%)	Y3 (%)
**X1:X2:X3**	0.1%:0.48%:16.32 mM	126 ± 8.70	91.06 ± 2.6	71.50 ± 6.5	121	93	72.21

## Data Availability

The data presented in this study are available in article.

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
