# Peer review of "Phytosterol-Loaded Surface-Tailored Bioactive-Polymer Nanoparticles for Cancer Treatment: Optimization, In Vitro Cell Viability, Antioxidant Activity, and Stability Studies"

_gels, 2022, doi:10.3390/gels8040219_

Round 1

Reviewer 1 Report

1. The reviewer would like to have more information about the toxicity of the Alginate/Chitosan-nanoparticle (β-SIT-Alg/Ch-NPs) to the normal cells. 

2. The reviewer would like to know how to apply in treatment e.g. in animal model. The bioavailability of the phytosterol in the nanoparticles and the free phytosterol in the animal model.

Author Response

Responses to reviewer 1 comments

Reviewer 1

Comments and Suggestions for Authors

  1. The reviewer would like to have more information about the toxicity of the Alginate/Chitosan-nanoparticle (β-SIT-Alg/Ch-NPs) to the normal cells.

Response: Authors agreed with the reviewer comment and it is quite relevant to investigate cytocompatibility against normal cells, albeit literature survey showed that the origin of Phytosterol is a natural phytoactive which are generally nontoxic to normal cells in the literature and regarded as safe. Other ingredients used in formulation were also approved and safe. Therefore, cytocompatibility against normal cells were not performed. Moreover, chitosan, a natural biodegradable and biocompatible polymeric carrier which is regarded as safe for human dietary use and approved for uses in biomedical and drug delivery application. For future work, authors will keep in mind these suggestions.

In support of the safety concern to human use and to the normal cells following literature has been quoted herein;

  1. Giovino C., Ayensu I., Tetteh J., Boateng J.S. Development and characterisation of chitosan films impregnated with insulin loaded PEG-b-PLA nanoparticles (NPs): A potential approach for buccal delivery of macromolecules. Int. J. Pharm. 2012;428:143–151. doi: 10.1016/j.ijpharm.2012.02.035.
  2. Xue M, Hu S, Lu Y, Zhang Y, Jiang X, An S, Guo Y, Zhou X, Hou H, Jiang C. Development of chitosan nanoparticles as drug delivery system for a prototype capsid inhibitor. Int J Pharm. 2015; 495(2):771-82.
  3. Rawal T, Parmar R, Tyagi RK, Butani S Colloids Surf B Biointerfaces. Rifampicin loaded chitosan nanoparticle dry powder presents an improved therapeutic approach for alveolar tuberculosis. 2017, 1; 154():321-330.
  4. Pistone S., Goycoolea F.M., Young A., Smistad G., Hiorth M. Formulation of polysaccharide-based nanoparticles for local administration into the oral cavity. Eur. J. Pharm. Biopharm. 2017;96:381–389.
  5. Chitosan in nasal delivery systems for therapeutic drugs. Casettari L, Illum L. J Control Release. 2014 Sep 28; 190():189-200.
  6. Chitosan nanoparticles are compatible with respiratory epithelial cells in vitro. Grenha A, Grainger CI, Dailey LA, Seijo B, Martin GP, Remuñán-López C, Forbes B. Eur J Pharm Sci. 2007; 31(2):73-84.
  7. The reviewer would like to know how to apply in treatment e.g. in animal model. The bioavailability of the phytosterol in the nanoparticles and the free phytosterol in the animal model.

Response: As per reviewer comment, for the measurement of bioavailability in the animal model, following plan has been proposed and to be carried in future; 

Response: A randomized single dose pharmacokinetic would have been planned. As per the study rats will be divided into into two groups I and group II. Each group will be containing six animals and will have to receive a single oral dose of 20 mg/kg from β-SIT-Alg/Ch-NPs and β-SIT suspension. At predetermined time point, 100 μl of blood will have been taken out from the tail vein of animal in heparinized/EDTA coated Eppendorf tubes, centrifuged at specified 3,000 rpm for 20 mins. Plasma sample would have been kept at -20 oC until the analysis. Further protein fraction in the plasma (100 μl) has been precipitated using organic solvent in a mixture of acetonitrile (0.5 ml) and ethyl acetate (1 ml). Post centrifugation of plasma sample, organic aliquot fraction will be dried under vacuum and residue reconstituted with ethanol (100 μl) and 10 μl would be injected in the HPLC for analysis and pharmacokinetic data will be analyzed.         

Thanks to reviewer for spending their valuable time and energy for evaluating our manuscript.

Reviewer 2 Report

1- why do the authors, use phytosterol for their work? what are the therapeutic effects of phytosterol in cancer therapy?

2- why the author didn't mention the drug loading efficiency as well as the encapsulation efficiency? this is an important factor for continuing the work.

3- why the author didn't perform the cell internalization/uptake study in order to confirm the efficient therapeutic effect of their nanoformulation?

4- why the author didn't use more period time for drug release profiles such as 72 h and 96 h? many studies consider 72 h and/or 96 h instead of 48 h.

5- the author mentioned in their title the "solid tumor treatment" but we didn't see any tumor studies/treatment in their work.

6- how do the authors confirm the cancer cell killing properties of their nanoformulation is due to apoptosis processes and not necrosis? the flow cytometry and/or RT PCR studies are necessary for this purpose.

7- where is the curve of the DLS test for size and charge properties? please present the relevant figure/curve obtained from the Malvern Zetasizer.

8- the charge of prepared nanoformulation (+ 25 mV) isn't enough to maintain their stability and prevent them from the aggregation process.  The author should present the shelf life studies for their proving the nanoformulation stability.

 9- What is the importance of thermal studies for the nanoformulation ingredients?

10- many β-SIT-Alg/Ch-NPs-FA nanoparticles DO NOT MATCH with which is presented in the TEM image scale bar. Most of them are smaller than 126 nm. For this purpose, we need the polydispersity curve obtained from Malvern Zetasizer.

Author Response

Response to reviewer 2 comments

Reviewer 2

Comments and Suggestions for Authors

1- why do the authors, use phytosterol for their work? What are the therapeutic effects of phytosterol in cancer therapy?

Response: Phyotsterol is an important plant constituent similar to cholesterol. It helps to reduce the cholesterol level in body limiting intestinal absorption and thus reduces blood LDL level and lowered cardiovascular ailments. In cancer growth inhibition, phytosterol seems to work through various mechanistic pathways including suppression of carcinogen production, reduction of cancer cell growth, angiogenesis, invasion and metastasis. It is promoting apoptosis of cancerous cells via lowering blood cholesterol levels. Phytosterol may also assist in improving the activity of antioxidant enzymes and thence help in reducing oxidative stress.

2- why the author didn't mention the drug loading efficiency as well as the encapsulation efficiency? this is an important factor for continuing the work.

Response: Authors appreciate for the reviewer comment. As per the comment, authors have reported encapsulation efficiency as % drug encapsulation as one of response shown in Table 2 for the various formulation. The optimized formulation estimated 91.06±2.6% and 6.0±0.52% % drug encapsulation and drug loading efficiency. The %drug loading incorporated in the main text. The %EE and %DL were calculated using following equation;

%EE= Total quantity of β-SIT- Total quantity of β-SIT in supernatant/Total quantity of β-SIT * 100

%DL= β-SIT encapsulated in β-SIT- Alg/Ch-NPs/ Total quantity of β-SIT-Alg/Ch-NPs-FA weight*100

3- why the author didn't perform the cell internalization/uptake study in order to confirm the efficient therapeutic effect of their nanoformulation?

Response: We appreciate the reviewer concern for this comment and at same time authors would like to point that the aim of the study was to optimize β-SIT- Alg/Ch-NPs formulation applying statistical design analysis, surface functionalization of nanocarrier and development of folate conjugated NPs, perform in vitro characterization including size distribution, surface potential, in vitro release, few activities (cytotoxic and anti-oxidant). Considering the suggestion authors have to take a note in the future study.        

4- why the author didn't use more period time for drug release profiles such as 72 h and 96 h? many studies consider 72 h and/or 96 h instead of 48 h.

Response: We appreciate the reviewer comment, drug release assessed for 48 h as per literature. Earlier studies reported that β-SIT release from sitosterol-chitosan nanocomplex for 48 hrs. At the end of 40 hrs 36.23%, 67.78%, 70.28% drug released corresponding to (pH 7.4, 6.5 and 5.5) from polymeric matrix of chitosan NPs. After 40 hrs, in next 8 hrs only a fraction drug release increased which is not significant statistically. Moreover, the drug release from NPs is controlled by aqueous medium penetration in the polymeric matrix, followed by hydration, matrix swelling, erosion of matrix gelatinous mass and thus, diffusion of dissolved drug into the medium. Apart from this, release rate was also governed by the physicochemical features of drug, dose, pH of surrounding medium and nature of polymer in which drug is entrapped.  

5- the author mentioned in their title the "solid tumor treatment" but we didn't see any tumor studies/treatment in their work.

Response: Authors agreed to the reviewer comment and appreciate them for asking the same and authors would like to point that the aim of the study was to optimize β-SIT- Alg/Ch-NPs formulation applying statistical design analysis, surface functionalization of nanocarrier and development of folate conjugated NPs, perform in vitro characterization including size distribution, surface potential, in vitro release, few activities (cytotoxic and anti-oxidant) and the suggested studies has to be considered for the future work.     

6- how do the authors confirm the cancer cell killing properties of their nanoformulation is due to apoptosis processes and not necrosis? the flow cytometry and/or RT PCR studies are necessary for this purpose.

Response: Authors agreed to the reviewer comment and appreciate them and keeping in mind these suggestions for the future work. 

7- where is the curve of the DLS test for size and charge properties? please present the relevant figure/curve obtained from the Malvern Zetasizer.

Response: As per reviewer comment authors presented particle size distribution curve (A) and zeta potential (B) as supplementary data sheet.

8- the charge of prepared nanoformulation (+ 25 mV) isn't enough to maintain their stability and prevent them from the aggregation process. The author should present the shelf life studies for their proving the nanoformulation stability.

Response: Authors agreed to the reviewer comment, and keeping in mind these suggestions for the future work. 

9- What is the importance of thermal studies for the nanoformulation ingredients?
Response: As per reviewer comment, thermal study reported in the current study, a differential scanning calorimetry (DSC). This study helps to precisely measure the melting point of drug sample and at the same time drug-excipient compatibility could be revealed. The significant change in the melting point of drug in the physical mixture or formulation as compared to drug alone may be attributed to their chemical interaction of drug with the added excipients.        

10- many β-SIT-Alg/Ch-NPs-FA nanoparticles DO NOT MATCH with which is presented in the TEM image scale bar. Most of them are smaller than 126 nm. For this purpose, we need the polydispersity curve obtained from Malvern Zetasizer.

Response: As per reviewer comment, the particle size distribution curve included in the supplementary data sheet.

Thanks to reviewer for spending their valuable time and energy for evaluating our manuscript.

Reviewer 3 Report

On request of Gels I reviewed the paper entitled “Phytosterol-loaded surface tailored bioactive polymer nanoparticles for solid tumor treatment: Optimization, in vitro cell viability, anti-oxidant activity, and stability studies” by S. Karim et al. The work deals with the preparation and characterization of sitosterol loaded NPs based on chitosan and alginate. In my opinion the topic is quite out the scope of the journal, however I leave the Editor this concern. The work has several linguistic and stylistic shortcomings, while the experimental method can be considered sufficient. The abstract and the introduction should be rearranged because the first is confusing while the second is little focused on the effects of the drug and the need to encapsulate it. Several grammar and editing mistakes are present (lines 26, 89,112,514,517, 560…, so I suggest that Authors have the text read by a native English speaker.

Specific comments:

  • The abstract is too long, the Author should respect the limits suggested by the journal which correspond to 200 words. The writing is quite confusing and with syntax and grammar mistakes. The Authors should reorganise the text and be more concise and clear, i.e. at line 31 the Authors speak about FT-IR and DSC experiments while before and after describe release studies. From line 21 to line 36, which is more than half abstract length, the Authors describe only background and methods, results are mentioned from line 39. In the title the Authors mentioned anti-oxidant activity but in the abstract this study is not reported.
  • The Authors forgot to include the reference numbers in the text.
  • The introduction begins in a not logical sequence: epidemiological data should be reported firstly.
  • Line 74, the Authors reported several studies dealing with targeted drug delivery systems but none citation was included, please add at least a review.
  • Line 78, the ref Akhater 2017 is not properly focused on the topic discussed in the text. Moreover, also the drawbacks of the EPR effects could be discussed, such as the effect of the enhancement of the hydrostatic pressure on active targeting.
  • Line 92, please add references that demonstrate the feasibility of what is expected.
  • Please add a sketch featuring the nanoparticles’ formation steps and of the chemical reaction between folic acid and chitosan.
  • In the introduction nothing is reported about the biological effects of sitosterol and the state of the art of its encapsulation attempts (PLGA, SLN, Silver NP). A brief comment for each formulation type already present in the literature, highlighting lights and shadows for each approach, should be included. Indeed only in this way it is possible to underscore the achievements reached by this study.
  • Please renumber the Tables according their appearance in the text.
  • Line 559, please describe the HPLC method in a more detailed way (instrument, mobile phase, flow rate, column) and add a representative chromatogram in the supplementary information.
  • Line 566, please add the formula of EE% and DL%. In addition in the subsection it is described the method for EE% detection but not for the DL% estimation, so please be more clear and precise.
  • The subsection following 4.6.1 should be 4.6.1.1., I suggest to delete the title at line 543 and change the number of the sections below consequently.
  • Line 588, from which company were the rats purchased? Nothing has been declared about the consent for the treatment of the animals, nor their characteristics.
  • Line 627, the Authors reported the concentrations in 10-50 µg/mL but in the figure 8 the values are reported in 10-50 µM. Which is the right one? Moreover, please specify in the text that the concentration applied are referred to sitosterol.
  • Line 640, to which ICH guideline did the Authors refer to? A reference should be also embedded.
  • Line 238, in the supplementary information file original representative size distribution images obtained by Zeta sizer should be included.
  • Figure 5, the caption is wrong, are they FT-IR spectra, aren’t they?
  • Line 339, Figure 7 not 9

Authors should pay particular attention to the numbers of tables and figures (line 339) and also to the captions (figure 5). In general, the work is very inaccurate at the level of writing and exposure. The work must be carefully reread and rewritten in order to be considered publishable. Important information is missing about the HPLC detection method of sitosterol, and the treatment of animals.

I suggest the Authors to present their work properly next time and I’ll be glad to reconsider it.

Round 2

Reviewer 2 Report

I emphasise that the authors remove the "solid tumor treatment" from their manuscript title.

Author Response

I emphasise that the authors remove the "solid tumor treatment" from their manuscript title.

Response: As per reviewer comment, it has been removed from the title.

We sincerely thanks to reviewer for spending their valuable time for evaluating our manuscript.

Reviewer 3 Report

The Authors have satisfied all my requests, no more comments are needed. 

Just one thing, I recommend a carefull check of measurement units: mL not ml and g (rat weight) not gm

Author Response

Just one thing, I recommend a carefull check of measurement units: mL not ml and g (rat weight) not gm

Response: As per reviewer comment, it has been corrected.

We sincerely thanks to reviewers for spending their valuable time for evaluating our manuscript.